# Enhancing Genetic Medicine: Rapid and Cost-Effective Molecular Diagnosis for a *GJB2* Founder Mutation for Hearing Impairment in Ghana

**DOI:** 10.3390/genes11020132

**Published:** 2020-01-27

**Authors:** Samuel M. Adadey, Edmond Tingang Wonkam, Elvis Twumasi Aboagye, Darius Quansah, Adwoa Asante-Poku, Osbourne Quaye, Geoffrey K. Amedofu, Gordon A. Awandare, Ambroise Wonkam

**Affiliations:** 1West African Centre for Cell Biology of Infectious Pathogens (WACCBIP), University of Ghana, Accra P. O. Box LG 54, Ghana; smadadey@st.ug.edu.gh (S.M.A.); atelvis45@gmail.com (E.T.A.); quansahdarius@gmail.com (D.Q.); AAsante-poku@noguchi.ug.edu.gh (A.A.-P.); oquaye@ug.edu.gh (O.Q.); gawandare@ug.edu.gh (G.A.A.); 2Division of Human Genetics, Faculty of Health Sciences, University of Cape Town, Cape Town 7925, South Africa; wonkamedmond@yahoo.fr; 3Bacteriology Department, Noguchi Memorial Institute for Medical Research, University of Ghana, Accra P.O. Box LG 581, Ghana; 4Department of Eye Ear Nose & Throat, School of Medical Sciences, Kwame Nkrumah University of Science and Technology, Kumasi AK-039-5028, Ghana; amedofugk@yahoo.com

**Keywords:** hearing impairment, *GJB2*-p.R143W, NciI-RFLP, rapid diagnostic test, Ghana

## Abstract

In Ghana, gap-junction protein β 2 (*GJB2*) variants account for about 25.9% of familial hearing impairment (HI) cases. The *GJB2*-p.Arg143Trp (NM_004004.6:c.427C>T/OMIM: 121011.0009/rs80338948) variant remains the most frequent variant associated with congenital HI in Ghana, but has not yet been investigated in clinical practice. We therefore sought to design a rapid and cost-effective test to detect this variant. We sampled 20 hearing-impaired and 10 normal hearing family members from 8 families segregating autosomal recessive non syndromic HI. In addition, a total of 111 unrelated isolated individuals with HI were selected, as well as 50 normal hearing control participants. A restriction fragment length polymorphism (RFLP) test was designed, using the restriction enzyme NciI optimized and validated with Sanger sequencing, for rapid genotyping of the common *GJB2*-p.Arg143Trp variant. All hearing-impaired participants from 7/8 families were homozygous positive for the *GJB2*-p.Arg143Trp mutation using the NciI*-*RFLP test, which was confirmed with Sanger sequencing. The investigation of 111 individuals with isolated non-syndromic HI that were previously Sanger sequenced found that the sensitivity of the *GJB2*-p.Arg143Trp NciI*-*RFLP testing was 100%. All the 50 control subjects with normal hearing were found to be negative for the variant. Although the test is extremely valuable, it is not 100% specific because it cannot differentiate between other mutations at the recognition site of the restriction enzyme. The *GJB2-*p.Arg143Trp NciI*-*RFLP-based diagnostic test had a high sensitivity for genotyping the most common *GJB2* pathogenic and founder variant (p.Arg143Trp) within the Ghanaian populations. We recommend the adoption and implementation of this test for hearing impairment genetic clinical investigations to complement the newborn hearing screening program in Ghana. The present study is a practical case scenario of enhancing genetic medicine in Africa.

## 1. Introduction

Globally, the most prevailing sensorineural disorder is hearing impairment (HI) [1], which accounts for about 466 million people worldwide [2]. According to the World Health Organization fact sheet, an estimate of 900 million people will be living with the condition by the year 2050 [2]. Over 119 genes [3] with more than 1000 mutations have been associated with hearing impairment of varied degrees in different populations [1]. Gap-junction protein β 2 (*GJB2*) and gap-junction protein β 6 (*GJB6*) are the most common genes associated with the condition globally, with high prevalence reported in the European and Asian populations. However, recent data including the use of mouse models has indicated that mutations in the coding region of the *GJB6* gene do not result in hearing impairment. The large genomic deletions in *GJB6*, especially *GJB6*-D13S1830, alters a *cis-*acting element and subsequently abolishes the expression of the *cis-GJB2* allele [4,5]. Thus, the *GJB6* gene itself plays no role in the development of hearing impairment but the surrounding sequences consisting of the *cis-*acting elements are responsible for the development of hearing impairment [5,6]. Nevertheless, in most African populations, *GJB2* and *GJB6* variants are rarely implicated in hearing impairment [7,8] with some *GJB2* cases found in Morocco [9,10], Sudan, and Kenya [11], yet an exceptionally high prevalence is found in Ghana [12,13,14]. Indeed, in Ghana, *GJB2* mutation (p.Arg143Trp) in the homozygous state accounts for 25.9% of cases in families segregating non-syndromic HI, as well as 7.9% of non-familial non-syndromic congenital HI cases (Adadey et al., 2019). This Ghanaian exception, in the African context, is predominantly due to a *GJB2* founder mutation (p.Arg143Trp), which was first reported in a village known as “the deaf village”, Adamorobe [13]. Adamorobe is a village located in the Eastern Region of Ghana and known to have a high hereditary hearing impairment incidence [15]. As of 2012, 41 people living with deafness were recorded among a population of 3500 in Adamorobe [16]. In this village, both the hearing and the deaf citizens interact and live together in one society.

The exceptionally high proportion of *GJB2* (p.Arg143Trp) variant in Ghana has created the need to develop a simple tool for testing in order to support appropriate informed counselling and planning for appropriate interventions. To develop molecular diagnostic tools for screening non-syndromic HI, there is a need for utilizing population and ethnic specific genetic markers due to the ethnically diverse nature of hearing impairment genes [17,18]. Recent clinical genetic testing efforts are centered around targeted genomic enrichment and/or massive parallel sequencing [18,19,20]. There are some efforts to develop polymerase chain reaction (PCR)-based diagnostic tools for screening for hearing impairment; however, most of these tools are in combination with DNA sequencing technologies [21,22,23], which are not easily implementable in low-income countries. To develop cheaper but effective diagnostic tools, mutations specific to populations have been considered, especially in populations where *GJB2* is prevalent. Specific genetic tests have been developed for carrier testing and prenatal diagnoses for *GJB2*-35delG variant in Caucasian populations [24,25]. In this study, we sought to design a restriction fragment length polymorphism test for *GJB2*-p.Arg143Trp genotyping in Ghana.

## 2. Materials and Methods

### 2.1. Ethical Approvals

The study was performed in accordance with the Declaration of Helsinki. Ethical approval for the study was obtained from the Noguchi Memorial Institute for Medical Research Institutional Review Board (NMIMR-IRB CPN 006/16-17) and the University of Cape Town’s Faculty of Health Sciences’ Human Research Ethics Committee (HREC 104/2018). Written and signed informed consent was obtained from all participants who were 21 years of age or older, and from parents or guardians in cases of minors, with verbal assent from participants, including permission to publish photographs. 

### 2.2. Study Participants

Congenital hearing-impaired patients were recruited from schools for the deaf and from the Adamorobe community following procedures reported previously [12]. Briefly, all participants’ details, as well as their personal and family histories, were obtained; medical records were reviewed by a medical geneticist and an ear, nose, and throat (ENT) specialist when possible; and relevant data were extracted, including three-generation pedigrees and perinatal histories, using a structured questionnaire to query possible environmental causes of hearing impairment. A general systemic and otological examination and audiological evaluation were performed, including a pure tone audiometric test, following the recommendation number 02/1 of the Bureau International d’Audiophonologie (BIAP), Belgium, to classify hearing levels [26,27]. The audiometric tests were conducted using KUDUwave portable audiometer (KUDUwave, Johannesburg, South Africa) in a quiet room. In bilateral octaves, the air conduction thresholds were from 250 H_Z_ through to 8000 H_Z_ and the bone conduction from 250 H_Z_ through to 4000 H_Z_. The pure tone average was determined using thresholds at 500, 1000, 2000, and 4000 H_Z_.

The study participants were categorized into three groups: (1) deaf community-based familial cases, (2) nation-wide isolated/non-familial cases, and (3) control individuals without a personal or family history of HI. The first group was made of families segregating HI, with at least two affected individuals and with evidence of non-environmental causes. In this group, 30 study participants from 8 families segregating hearing impairment were recruited from the Adamorobe community in the Eastern Region of Ghana. Out of the 30 participants, 20 were hearing-impaired and 10 participants had normal hearing. Apart from the families with putative genetic etiology of hearing loss, an additional family was found to have a putative environmental etiology of the condition and was excluded from the study. The second group of participants was made up of 111 isolated/non-familial cases of unrelated probands with putative genetic causes of hearing impairment and were recruited from 6 schools for the deaf across Ghana. All the affected individuals (familial and isolated cases) considered for the study had congenital non syndromic HI. The third group (the control group) was made of 50 normal hearing participants that were randomly recruited nationwide from the Ghanaian population. 

### 2.3. Molecular Analyses

DNA extraction: Venous blood was collected from each participant and DNA was extracted from the blood samples using a QIAamp DNA Blood Maxi Kit (Qiagen, Germantown, MD, USA) in the Laboratory of West African Centre for Cell Biology of Infectious Pathogens (WACCBIP), University of Ghana, Accra, Ghana. 

Polymerase chain reaction (PCR) and Sanger sequencing: At the Division of Human Genetics, University of Cape Town, specific primers (Appendix A) were used to amplify the coding regions of *GJB2* (exon 2) and *GJB6*, as described by Bosch et al. in 2014. The annealing and extension temperatures for the PCR were 60 °C and 70 °C for 30 s and 1 min, respectively. The PCR amplicons were Sanger sequenced as described by Bosch et al. [28] using an ABI 3130XL Genetic Analyzer (Applied Biosystems, Foster City, CA, USA). Screening for del(*GJB6*-D13S1830) was performed as previously described, using primers and methods by del Castillo et al. [29]. 

Restriction fragment length polymorphism (RFLP) technique: The p.Arg143Trp variant in the *GJB2* gene was investigated using RFLP technique designed as follows. *GJB2*-specific primers [28] were used to amplify exon 2 of the gene where the p.Arg143Trp variant is located. Carefully selected restriction enzyme NciI (supplied by New England Biolabs Inc., Massachusetts, MA, USA, through Inqaba biotec, Pretoria, South Africa) with the recognition site “CCSGG” was used to digest the PCR amplicons. The gene layout and the *GJB2*-p.Arg143Trp NciI*-*RFLP design including the cut sites is illustrated in Figure 1. The RFLP reaction consisted of 15 μL of the PCR product, 2 μL of 10X buffer, 0.25 μL of an NciI enzyme (20,000 units/mL), and 2.75 μL of nuclease-free water. The restriction reaction mixture was incubated overnight at 37 °C. The digested products were resolved on 2% agarose gel for 1.5 h. The accuracy, sensitivity, and specificity of the RFLP test was determined as described by Baratloo et al. [30] using sequencing as the gold standard.

### 2.4. Data Analysis

Data from the study was inputted into Microsoft Excel and analyzed with GraphPad Prism version 6. One-way analysis of variance (ANOVA) was used to determine the differences between the mean hearing measurements (pure tone average) of the different *GJB2*-p.Arg143Trp genotypes. Tukey’s multiple comparisons test was used to compare between the *GJB2*-p.Arg143Trp genotypes. The specificity and sensitivity of the RFLP test were calculated as described by Schrauwen et al. [23].

## 3. Results

### 3.1. Selected Families Segregating Hearing Impairment from Adamorobe Village, Ghana

In this study, 8 families from Adamorobe were found to have 2 or more family members living with hearing impairment (Figure 2), from which 20 congenital deaf and 10 normal hearing family members were identified. Audiological assessment of the participants from Adamorobe revealed that all the hearing-impaired patients had profound sensorineural HI. The unaffected family members without the homozygous mutant (TT) genotype had normal-to-moderate hearing impairment.

### 3.2. Restriction Fragment Polymorphism Design for GJB2-p.Arg143Trp

The target region of *GJB2-*p.Arg143Trp variant was PCR amplified for each participant (Appendix A). The NciI restriction enzyme had two restriction sites on the DNA amplified (Figure 1A) and cleaves the PCR amplicons of the wildtype (CC genotype) samples to give three products of the lengths 527, 123, and 155 bp. The NciI restriction digest of the homozygous mutant (TT) produced two fragments of the lengths 600 and 155 bp, with the enzyme cutting only once. The heterozygous carriers (CT) yielded four different fragments (600, 527, 123, and 155 bp) (Figure 1B). The NciI enzyme cleaved the PCR product in any of the above circumstances; this served as an internal control, and hence an invalid test was when there was no cleavage. The *GJB2*-p.Arg143Trp NciI*-*RFLP genotyping results of 20 selected samples from Adamorobe were validated using Sanger sequencing (Figure 3). 

### 3.3. GJB2-p.Arg143Trp NciI-Restriction Fragment Polymorphism Investigations

The molecular analysis using *GJB2*-p.Arg143Trp NciI*-*RFLP test identified 18 out of the 20 hearing-impaired patients, from 7/8 families, to be homozygous for the p.Arg143Trp (TT) variant. In the eighth family were two individuals affected by HI, one was heterozygous (CT) and the other had the CC genotype (Figure 2B). In order to exclude *GJB6-*related HI in this family, we investigated variants in *GJB6*, and no variant was found. No other participant had a variant in the *GJB6* gene, (*n* = 20).

Seven (7) out of the 10 family members without hearing impairment were heterozygous (CT), thus having the p.Arg143Arg/p.Arg143Trp variant, while the rest had the p.Arg143 variant (Figure 3A). 

A total of 111 individuals with non-familial isolated non-syndromic HI, whose samples were previously Sanger sequenced for *GJB2* variants [12], were analyzed using the developed *GJB2*-p.Arg143Trp NciI*-*RFLP test. Table 1 illustrates that the *GJB2*-p.Arg143Trp NciI*-*RFLP test was found to have 100% sensitivity compared to Sanger sequencing as the gold standard. To examine the clinical applicability of the test, 50 control participants with normal hearing were screened and found negative for the *GJB2*-p.Arg143Trp variant.

### 3.4. Genotype to Phenotype Correlations

On the basis of *GJB2-*p.Arg143Trp genotypic classification of the familial cases from Adamorobe, the pure tone average of homozygous mutant (TT) ranged from 97 to 108 dB with a mean of 105.4 and 107.3 dB in the left and right ears, respectively. The pure tone average range for the heterozygote (CT) was from 17 to 108 dB, with a mean of 43.6 and 40.6 dB in the left and right ears, respectively. The range for the homozygous CC genotype (p.Arg143Arg) was from 18 to 108 dB, with the mean 53 and 46.5 dB in the left and right ears, respectively. There was a statistically significant difference between the audiometric measurements of the TT and CT genotypes in both ears. Similarly, in both ears, there was a statistically significant difference between the TT and CC genotypes (Figure 4). 

## 4. Discussion

This study designed a restriction fragment length polymorphism (RFLP) test for effective screening of *GJB2*-p.Arg143Trp (rs80338948). The *GJB2*-p.Arg143Trp variant results from a pathogenic point mutation (c.427C > T) in the exon 2 of the connexin 26 gene on chromosome 13 [13,14]. The drive for an efficient and cost-effective test was from the fact that the founder mutation, *GJB2*-p.Arg143Trp, is the most common variant associated with hearing impairment in Ghana [12,13,14]. 

The use of next generation sequencing (NGS) has been proposed as the best tool for the discovery of hearing impairment genes [32], especially in Africa because of the high diversity within the African population [33,34]. Due to ethical and social challenges, NGS needs to be carefully considered in clinical practice [35]. In developing countries, the clinical use of NGS is still a major challenge because of the associated high cost of the equipment and the computational challenges posed by the approach [36]. However, there were some attempts to develop relatively simple, low cost, and population-specific screening approaches for some of the major hearing impairment gene mutations [37,38,39]. 

For the first time, we designed and tested the effectiveness of RFLP, using the NciI enzyme, to screen for the founder mutation (*GJB2*-p.Arg143Trp) in Ghana. Accuracy, sensitivity, specificity, and predictive values are critical parameters considered for the clinical use of a test [30,40]. Our *GJB2*-p.Arg143Trp NciI*-*RFLP test had good positive and negative predictive values for genotyping of the *GJB2*-p.Arg143Trp variant in the Adamorobe participants from Ghana, and also in a nationwide sample of unrelated affected individuals. Nevertheless, the test cannot differentiate between other variants within the recognition site of the restriction enzyme; hence, similar results would be obtained for the following pathogenic mutations: p.Phe142Leu (c.426C > A), p.Y142del (c.424_426delTTC), and p.Arg143Gln (c.428G > A). To confirm the specific mutation at the enzyme restriction site, Sanger sequencing would be needed. However, the high prevalence of the *GJB2*-p.Arg143Trp variant within the Ghanaian population makes the NciI*-*RFLP test relevant. A 100% sensitivity was obtained for the *GJB2*-p.Arg143Trp NciI*-*RFLP test when compared with the gold standard, Sanger sequencing. Although a single gene test for hearing impairment is inefficient for many populations [39], the aforementioned qualities of the test would enable it to be used as a first-line diagnosis for hearing impairment genetics in the newborn hearing screening program, as well as for prenatal testing. The *GJB2*-p.Arg143Trp NciI*-*RFLP test would therefore be of great clinical value in Ghana. 

The *GJB2*-p.Arg143Trp NciI*-*RFLP test identified the founder mutation in the eight Adamorobe families investigated. In all the families, the mutation segregated with the phenotype, and all affected individuals reported a homozygous variant (TT genotype), except in one family where one affected individual was heterozygous (CT) and the other without any variant (CC), suggesting that there are other genes still to be discovered to explain the HI in this family. Similar to a family from Japan [37], the heterozygous *GJB2*-p.Arg143Trp variant in the above family did not segregate with the HI phenotype (Figure 2B). Variants in the *GJB6* gene are no longer considered as causes of hearing impairment. However, the presence of *GJB6* variants affecting the *cis-*acting element upstream of both *GJB6* and *GJB2* in association with variants in *GJB2* (digenic inheritance) are now known to be pathogenic through the modification of *GJB2* expression [5,6]. Hence, we sought to exclude any *GJB6* variant that might disrupt the *cis-*acting element. We therefore investigated *GJB6* variants in particular; *GJB6*-D13S1830 and no *GJB6* pathogenic variant was identified in this family. Hence, we propose the use of whole exome sequencing (WES) in future, or targeted panel sequencing, which has been shown to be efficient in Cameroonian families [33], to further investigate this family, as well as any other family that is specifically negative for the *GJB2*-p.Arg143Trp variant in Ghana. 

*GJB2*-p.Arg143Trp variant is known to be associated with profound HI [13,14,37]. The audiometric characterization of the *GJB2*-p.Arg143Trp homozygous individuals showed that they had profound HI. Previous studies by Brobby et al. from the same village indicated that *GJB2*-p.Arg143Trp homozygous individuals express profound hearing impairment [13]. Similar to the previous report [13], we found that there was no significant difference between the average hearing levels of the CT (heterozygote for the pathogenic variant) and the CC (non-carrier of the pathogenic variant) genotypes. Our results and previous reports confirmed the autosomal recessive mode of inheritance of *GJB2*-p.Arg143Trp [12,13,14]. 

## 5. Conclusions

We developed a rapid and cost-effective NciI-RFLP test for the *GJB2*-p.Arg143Trp founder mutation in Ghana. The *GJB2*-p.Arg143Trp NciI*-*RFLP test had 100% sensitivity when compared with Sanger sequencing, the gold standard. We therefore propose that testing for *GJB2*-p.Arg143Trp variant using the NciI-RFLP test should be implemented as part of the newborn hearing screening program in Ghana, a practical case scenario of enhancing genetic medicine in Africa. 

## Figures and Tables

**Figure 1 genes-11-00132-f001:**
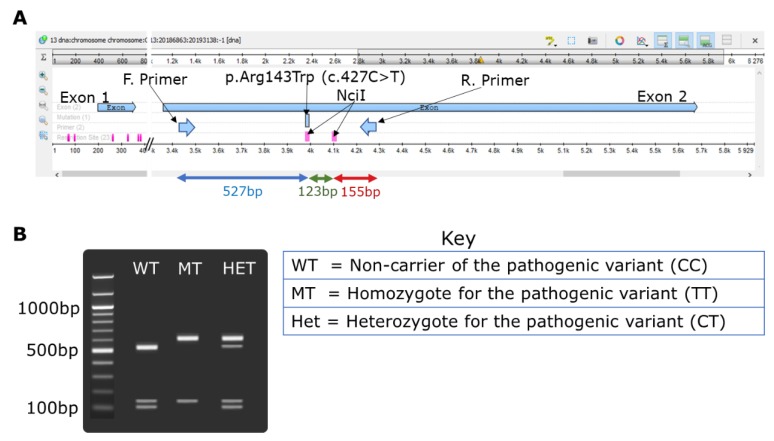
NciI restriction fragment polymorphism investigations for gap-junction protein β 2 (*GJB2*)-p.Arg143Trp (c.427C > T rs80338948) variant. (**A**) Unipro UGENE [31] map of *GJB2* exon 2 showing the primer binding sites (F. primer and R. primer) and the restriction sites (CCSGG) for the restriction enzyme NciI and the resulting DNA fragments. (**B**) Expected gel electrophoresis result.

**Figure 2 genes-11-00132-f002:**
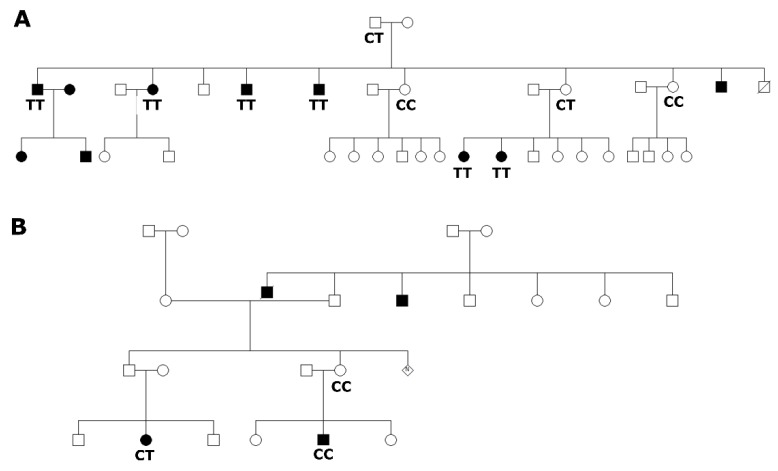
Pedigrees and genotypes of familial cases from Adamorobe. (**A**) Representative pedigree of families that segregate *GJB2*-p.Arg143Trp (c.427C > T rs80338948) variant with hearing impairment. (**B**) Pedigree of a family that did not segregate *GJB2*-p.Arg143Trp variant with the phenotype.

**Figure 3 genes-11-00132-f003:**
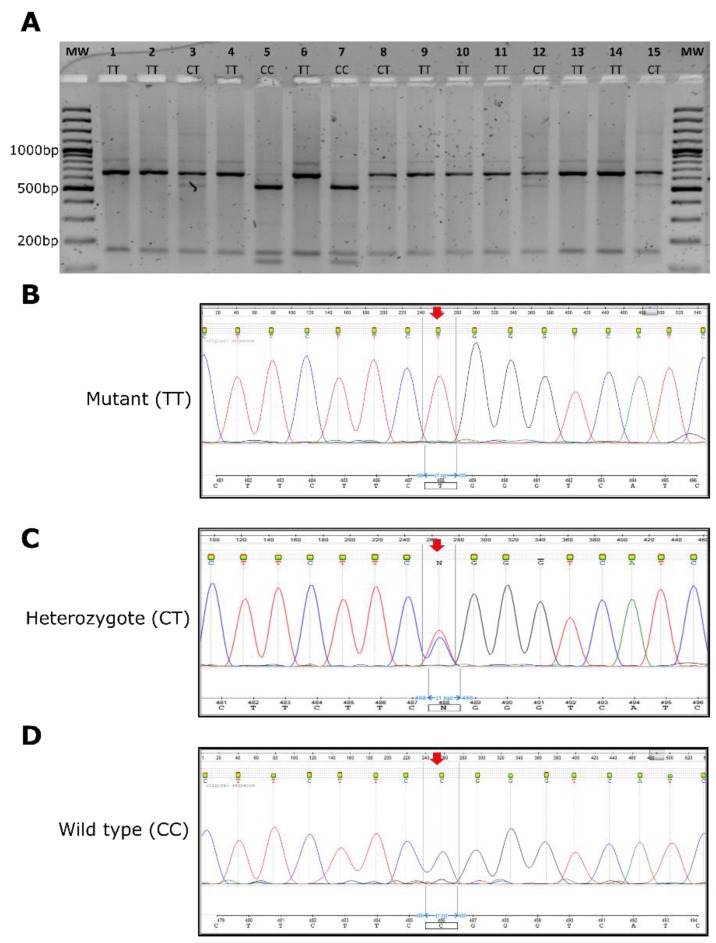
*GJB2*-p.Arg143Trp screening. (**A**) Representative gel of NciI*-*restriction fragment polymorphism (RFLP) test used to screen samples for *GJB2*-p.Arg143Trp variant. (**B**–**D**) Representative chromatograms of Sanger sequences validating the p.Arg143Trp NciI*-*RFLP results.

**Figure 4 genes-11-00132-f004:**
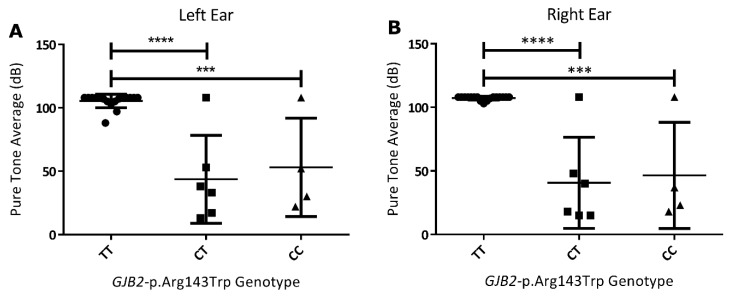
Audiological characterization of hearing-impaired participants from the deaf community of Adamorobe. (**A**) Left ear and (**B**) right ear pure tone average of participants according to their *GJB2*-p.Arg143Trp genotypes. The age range of the genotypes TT (*n* = 17), CT (*n* = 6), and CC (*n* = 4) were 9 to 80 years, 23 to 66 years, and 11 to 63 years, respectively. *p*-values less than 0.05 were considered significant. *p*-values less than 0.0001 and 0.001 are represented by (****) and (***), respectively.

**Table 1 genes-11-00132-t001:** Validation of *GJB2*-p.Arg143Trp NciI*-*restriction fragment polymorphism tests with Sanger sequencing.

**Familial Cases from Adamorobe**				
		Sanger Sequencing		
	Genotype	TT	CT	CC
*GJB2*-p.Arg143Trp NciI*-*RFLP	TT	12	0	0
	CT	0	6	0
	CC	0	0	2
**Nation-Wide Isolated/Non-Familial Cases**				
		Sanger Sequencing		
	Genotype	TT	CT	CC
*GJB2*-p.Arg143Trp NciI*-*RFLP	TT	6	0	0
	CT	0	1	0
	CC	0	0	104

The mutant, heterozygote, and wild type are represented by TT, CT, and CC, respectively.

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
