# Peer review of "Enhancing Genetic Medicine: Rapid and Cost-Effective Molecular Diagnosis for a GJB2 Founder Mutation for Hearing Impairment in Ghana"

_genes, 2020, doi:10.3390/genes11020132_

Round 1

Reviewer 1 Report

The m/s concerns the development of a screening test for a high prevalence autosomal recessive GJB2 mutation in a Ghanaian population. It is really encouraging to see the use of genetic screening technology being applied in an African context. I applaud the authors for their efforts to improve the diagnosis of hearing loss in this underserved population. In general the m/s reads very well and most of my points are minor. These are listed below. The one major issue is the description of the samples tested. Can the authors more clearly outline their study participants in the methods section? First, state the three groups that were targeted then give a brief description of n size, how sampling was conducted, etc., for each group. For example, at present the reader does find out the normal hearing group was a nationwide sample until the discussion section of the m/s. 

Minor concerns

Title: For consistency it is better to capitalise "global" -- i.e., write "Global".

Abstract: A few words are missing - write "... but has not yet been investigated ..."; throughout the m/s replace "new-born" with "newborn" - this is now the much more usual spelling.

Introduction: page 2, line 48 should be "non-syndromic HI"; line 56 better if "... in Ghana creates the need ..."; lines 58-60 This sentence, commencing "The development ..." is confusing and I'm not quite sure what you mean. Please rephrase; line 62 write the term for "PCR" in full and then "... (PCR) based ..." In the methods section you can then just write "PCR".

Materials and Methods: line 71, to be consistent with other headings this should be "Ethical approvals"; line 86 should be "... were performed, including a pure tone audiometric test ..."; page 3, line 98 to be consistent with other headings this should be "Molecular analyses"; line 106 "Bosch et al. in 2014" -- incorrect referencing style. This needs to become reference [25] and other references modified accordingly. Add details of the reference to the reference list; page 4, line 128-129 same problem with "Schrauwen et al" - at present not in references.

Results: line 131 should be "... hearing impairment from ..."; line 150 Figure 3 title is shown but FIGURE IS MISSING from my version of the m/s; page 5, line 172 there needs to be a Note at the bottom of Table 1 giving the full names for TT, CT and CC; page 6, line 183 should be "...and (B) right ear ...".

Discussion: line 194 better if "... has been proposed as the best tool ...". No need for "and used" because usage and its problems are mentioned later; line 205 should be "... test had good positive ..."; line 206 should be "... in a nationwide sample of unrelated ..."; line 208 should be "Although a single ..."; line 211 delete "(NHS)" because the abbreviation is not used later in the m/s (therefore redundant); page 7 line 227 "Broddy et al, 1998" another example of wrong referencing style (and also incorrect spelling) -- are you referring to reference [10]?; line 229 better if "... confirmed the autosomal ..".

References: [5] delete the redundant second title -- just need "South African medical journal"; [12] give more information - should be "Utrecht, LOT Publications..."; [28] typo "...the _UGENE_team", delete the _; [37] should be "eJIFCC" 

Author Response

Reviewer 1.

Reviewer’s comment:

The m/s concerns the development of a screening test for a high prevalence autosomal recessive GJB2 mutation in a Ghanaian population. It is really encouraging to see the use of genetic screening technology being applied in an African context. I applaud the authors for their efforts to improve the diagnosis of hearing loss in this underserved population. In general, the m/s reads very well and most of my points are minor. These are listed below. The one major issue is the description of the samples tested. Can the authors more clearly outline their study participants in the methods section? First, state the three groups that were targeted then give a brief description of n size, how sampling was conducted, etc., for each group. For example, at present the reader does find out the normal hearing group was a nationwide sample until the discussion section of the m/s. 

Authors response:

Thank you for your positive comments. We have refined the description of the study population as follows:

The study participants were categorized into three groups: 1) deaf community based familial cases, 2) nation-wide isolated/non-familial cases, and 3) controls individual without personal or familial history of HI. The first group was made of families segregating HI, with at least two affected individuals and with strong evidence of non-environmental causes. In this group, 30 study participants from 8 families segregating hearing impairment were recruited from Adamorobe community in the Eastern Region of Ghana. Out of the 30 participants, 20 were hearing impaired and 10 participants had normal hearing. Apart from the families with putative genetic etiology of hearing loss, an additional family was found to have putative environmental etiology of the condition and was excluded from the study. The second group of participants that was made of 111 isolated/non-familial cases of unrelated probands with putative genetic causes of hearing impairment and were recruited from 6 schools for the deaf across Ghana. All the affected individuals (familial and isolated cases) considered for the study had congenital non syndromic HI. The third group, the control group, was made of 50 normal hearing participants that were randomly recruited nationwide from the Ghanaian population.

Minor concerns

Reviewer’s comment:

Title: For consistency it is better to capitalise "global" -- i.e., write "Global".

Authors response:

The world “Global has been deleted as recommended by the second reviewer.

Reviewer’s comment:

Abstract: A few words are missing - write "... but has not yet been investigated ..."; throughout the m/s replace "new-born" with "newborn" - this is now the much more usual spelling.

Authors response:

The abstract has been revised: the missing words have been inserted and “new-born” changed to “newborn”.

Reviewer’s comment: Introduction: page 2, line 48 should be "non-syndromic HI"; line 56 better if "... in Ghana creates the need ..."; lines 58-60 This sentence, commencing "The development ..." is confusing and I'm not quite sure what you mean. Please rephrase; line 62 write the term for "PCR" in full and then "... (PCR) based ..." In the methods section you can then just write "PCR".

Authors response:

Line 56 has been edited accordingly

Line 58-60 has been rephrased as “To develop molecular diagnostic tools for screening non-syndromic hearing impairment, there is a need for utilizing population and ethnic specific genetic markers due the ethnically diverse nature of hearing impairment genes.”

Line 62 has been rephrased as “There are some efforts to develop polymerase chain reaction (PCR) based diagnostic tools for screening hearing impairment, however, most of these tools are in combination with”

Reviewer’s comment:

Materials and Methods: line 71, to be consistent with other headings this should be "Ethical approvals"; line 86 should be "... were performed, including a pure tone audiometric test ..."; page 3, line 98 to be consistent with other headings this should be "Molecular analyses"; line 106 "Bosch et al. in 2014" -- incorrect referencing style. This needs to become reference [25] and other references modified accordingly. Add details of the reference to the reference list; page 4, line 128-129 same problem with "Schrauwen et al" - at present not in references.

Authors response:

Line 71 corrected to “Ethical approvals” and line 98 corrected to “Molecular analysis”

Line 86 has been corrected as “A general systemic and otological examination and audiological evaluation were performed including a pure tone audiometric test …”

Line 106 has been edited with the correct insertion of the references.

The reference on line 128-129 has been inserted.

Reviewer’s comment:

Results: line 131 should be "... hearing impairment from ..."; line 150 Figure 3 title is shown but FIGURE IS MISSING from my version of the m/s; page 5, line 172 there needs to be a Note at the bottom of Table 1 giving the full names for TT, CT and CC; page 6, line 183 should be "...and (B) right ear ...".

Authors response:

Line 131 corrected as “In this study, 8 families from Adamorobe were found to have two or more family members living with hearing impairment (Figure 2) from which 20 deaf and 10 normal hearing family members were identified”

Line 150, has the figure on the MS word version.

The Table 1 has “The mutant, heterozygote and wild type are represented by TT, CT and CC respectively” added to the bottom of the table.

Line 183 corrected as “… Left ear and (B) right Ear pure…”

Reviewer’s comment:

Discussion: line 194 better if "... has been proposed as the best tool ...". No need for "and used" because usage and its problems are mentioned later; line 205 should be "... test had good positive ..."; line 206 should be "... in a nationwide sample of unrelated ..."; line 208 should be "Although a single ..."; line 211 delete "(NHS)" because the abbreviation is not used later in the m/s (therefore redundant); page 7 line 227 "Broddy et al, 1998" another example of wrong referencing style (and also incorrect spelling) -- are you referring to reference [10]?; line 229 better if "... confirmed the autosomal ..".

Authors response:

Line 194 corrected as “The use of next generation sequencing (NGS) has been proposed as the best tool for the discovery of hearing impairment genes”

Line 206 has been revised as “, and also in a nationwide sample of unrelated affected individuals”

line 211: the abbreviation "(NHS)" has been deleted

Line 227: appropriated reference format has been inserted

Line 229: the phrase “autosomal recessive” is confirmed.

Reviewer 2 Report

This manuscript reports on a RFLP test to detect the p.Arg1434Trp mutation in GJB2, which is a common cause of non-syndromic hearing impairment in the population of Ghana.

Major comment

The manuscript describes a quick, cheap and easy method for the detection of a clinically relevant mutation. Additional details will improve the description. The NciI recognition sequence should be shown, and the nucleotide position that is affected by the mutation should be indicated. Note that the mutation destroys the recognition site and so the test is not theoretically specific. Any other change in the recognition site will be detected as well. For example, the c.426C>A change (resulting in the pathogenic variant p.Phe142Leu) would result in the same restriction pattern as that of p.Arg143Trp. Although the test will still be useful for genetic screening in Ghana, given the high frequency of p.Arg143Trp in this population, this issue should be discussed in the text, and it should be kept in mind when performing the screenings, i.e. Sanger confirmation would still be needed.

Minor points

In my opinion, the title is too long and the first sentence is a little pretentious. I would delete “Enhancing global Genetic Medicine:”. Line 81. “reported previously reported” (a typo) Line 99. “vinous blood” should read “venous blood” (a typo). Line 136. Did UNAFFECTED family members have moderate hearing impairment? I suppose that the authors mean that family members without the homozygous genotype had normal to moderate hearing impairment. Line 226. “profound severity of HI” should read “profound HI”.

Author Response

Reviewer 2

Reviewer’s comment:

Several recessive pathogenic variants in GJB2 have been associated with profound hearing loss in Caucasian, Ashkenazi Jewish and East Asian populations. The presence of biallelic pathogenic variants in GJB2 is often a frequent cause of hearing loss in these populations. In Africa, this does not seem to be the case, except in Ghana where the pathogenic variant p.Arg143Trp in GJB2 is frequent. In this manuscript, the authors describe the design of a test, and its validation in 2 cohorts of patients with hearing loss and their associated controls, to identify the presence of the variant p.Arg143Trp in GJB2, which is frequent in Ghana. This test is based on a PCR targeting GJB2 only coding exon (exon 2), NciI restriction enzyme digestion of the PCR products, followed by gel electrophoresis and analysis. This variant p.Arg143Trp is pathogenic and when present as homozygote is associated with profound hearing loss. The authors show that this test is specific and sensitive and could thus be a very useful clinical test, which could be more easily implemented than classical sequencing approaches, as it does not require the use of a sequencer. The authors propose for this test to be done in addition to the new-born hearing screening program in Ghana.

The rationale for this study is strong.  The writing is generally clear although a few sentences require clarifications.  Some important updates regarding the role of GJB6 in hearing loss are needed. Although the authors’ conclusions are well supported by the data, presentation of a negative PCR control in figure 3 would be helpful (see below). Some clarifications are also needed relative to their hearing tests and the presentation of their audiometric results.

The implementation of the simple and robust test described in this work, in the clinical practice, could be very impactful for the early diagnosis of the genetic cause of hearing loss in Ghana.

Authors response:

Thank you for your positive comments.

Major comments:

Reviewer’s comment:

Nomenclature of the variant should be clearly indicated at both the DNA and the protein level; reference accession numbers should be indicated.

Authors response:

Reference accession number has been added “...GJB2-p.Arg143Trp  (NM_004004.6:c.427C>T/rs80338948)...”.

Reviewer’s comment:

The association of hearing loss and the presence of biallelic pathogenic variants in the coding sequence of GJB6 or of one variant in GJB6 coding sequence in association with a mutation in GJB2 (digenic inheritance), has been refuted (see Genet Med.2019 Oct;21(10):2239-2247. doi: 10.1038/s41436-019-0487-0. ClinGen expert clinical validity curation of 164 hearing loss gene-disease pairs. DiStefano MT et al. ClinGen Hearing Loss Clinical Domain Working Group.)

Please modify the text accordingly.

Deletions in the region of GJB2/GJB6, which are 2 small genes close to each other, are thought to be pathogenic through the modification of GJB2 expression. 

Authors response:

Thanks very much for this important update. We have amended our text in the introduction and discussion, accordingly.

Reviewer’s comment:

Presentation of a negative PCR control in figure 3 would be appreciated unless the authors prefer presenting the PCR results, they initially obtained (with a negative control), before presenting their digestion results. Please present the results of Figure 3 as you suggest for the results to be presented in clinic. So that this article contains all the information necessary to do the test proposed, wouldn’t it be worth indicating the PCR conditions and the sequences of the primers used as template, still referencing the initial article those were first reported in as you did?

Authors response:

The PCR results initially obtained (with a negative control), before the restriction digestion is added as Figure S1 and the primer sequences are added as Table S1. The PCR conditions are added “The annealing and extension temperatures for the PCR were 60˚C and 70˚C for 30 seconds and 1 minute respectively”.

Reviewer’s comment:

Please indicate clearly the results of which participants are included in Figure 4. Are these audiograms those of the 20 participants with hearing loss and 10 normal hearing family members from Adamorobe?

Authors response:

The revision has been made and the participants for the results on Figure 4 stated: “Based on GJB2-p.Arg143Trp genotypic classification of the familial cases from Adamorobe,..”

Reviewer’s comment:

Please describe the material and method used to obtain the audiological data you present. Please explain what the pure tone average represents and the set of frequencies you chose to test and include. This should be explained in the text.

Authors response:

The text has been revised in the method accordingly: “The audiometric tests were conducted using KUDUwave™ portable, audiometer (KUDUwave™, Johannesburg, South Africa) in a quiet room. In bilateral octaves, the air conduction thresholds were from 250HZ through to 8000HZ and the bone conduction from 250HZ through to 4000HZ. The pure tone average was determined using thresholds at 500HZ, 1000HZ, 2000HZ and 4000HZ.”

Reviewer’s comment:

The pure tone average of the participants which carry the genotype TT are very high. Did you obtain some positive response in most of those patients? Is the material used capable of reliable detection of these high thresholds in the conditions of testing? 

Is there a difference in hearing thresholds between both ears in participants of each genotype?

Would it be better to present the data based on best ear and/or worse ear rather than left and right ear or even indicate both ears on the same graph and show the results at the different frequencies tested as a function of the genotype.

How different are those results as compared to those presented by G.W Brobby et al. 1998? Please comment

Authors response:

The equipment used is capable of measuring responses at the reported thresholds and some positive responses were obtained from some of the patients.

There were some slight differences in the hearing threshold between both ears for each genotype hence the Figure 4 had both ears represented. The differences were however statistically not significant.

Our result is comparable to the previous report from Brobby et al., and the manuscript has been revised in the discussion to reflect that as follows:  “The audiometric characterization of the GJB2-p.Arg143Trp homozygous individuals showed that they had profound severity of HI. Previous studies by Brobby et al, from the same village indicated that GJB2-p.Arg143Trp homozygous individuals express profound hearing impairment [10]. Similar to the previous report [10], we found that there was no significant difference between the average hearing levels of the CT (carrier) and the CC (unaffected) genotypes.”

Reviewer’s comment:

Please indicate in the legend of Figure 4 how many participants were tested in each group presented (TT, CT, CC). Please indicate their age or age range.

No comment is made in the manuscript regarding the age of the participants of each group. Do the patients with TT genotype present profound hearing loss from birth or is their hearing loss progressive?

Authors response:

Legend of Figure 4 has been updated as “The age range of the genotypes TT (n=17), CT (n=6) and CC (n=4) were 9 to 80 years, 23 to 66 years and 11 to 63 years respectively.”

The following statement were updated to indicate that the hearing impaired participants had congenital hearing loss.

Page 2 under the study participant section: “Congenital hearing impaired patients were recruited from schools for the deaf and Adamorobe community following procedures reported previously reported” Page 2 under the study participant section: “All the affected individuals considered for the study had congenital HI”.

Other comments:

Reviewer’s comment:

The authors may want to replace along the manuscript “hearing impairment” by “hearing loss”, the preferred term in the field.

Authors response:

Due to the stigmatization associated with the condition, most deaf people in Ghana and the deaf community prefer the term “hearing impairment” to “hearing loss”. We made the above observation during our field work and found that it was consistent with previous publications from Ghana (Brobby et al., DOI: 10.1056/NEJM199802193380813 and Hamelmann et al., DOI: 10.1002/humu.1156). Based on the above reasons we prefer to use the term which is well accepted by our cohort. In addition, a recent publication on the hearing impairment ontology suggests hearing impairment as a standardized term (Hotchkiss et all 2019, DOI:10.3390/genes10120960 ).

Reviewer’s comment:

The use of the term “global” here along the manuscript seems inappropriate. As pointed out by the authors, the variant studied here is only found at a high frequency in Ghana and not in other countries of Africa. I would suggest taking this term out but keeping the rest of the sentences identical (lines 2, 35, 237).

Authors response:

The term “global has been deleted from the respective lines.

Reviewer’s comment:

Please include information on prevalence of GJB2 mutation in Ghana in the abstract, which support the importance of the diagnostic test described.

MIM number should be included in the text.

Line 18, “major” should be replaced by “most frequent”

Authors response:

Revision made in text as follows “In Ghana, GJB2 mutations account about 25.9% of familial hearing impairment (HI) cases of which GJB2-p.Arg143Trp (NM_004004.6:c.427C>T/OMIM: 121011.0009/rs80338948) founder variant remains the most frequent mutation associated with congenital HI, but has not yet been investigated in clinical practice.”

Reviewer’s comment:

Lines 58 to 61: please rephrase sentence

Authors response:

The sentence has been rephrased as “To develop molecular diagnostic tools for screening non-syndromic hearing impairment, there is a need for utilizing population and ethnic specific genetic markers due the ethnically diverse nature of hearing impairment genes”

Reviewer’s comment:

Line 64: you may want to change “applicable to” to “implementable in”

Authors response:

The word implementable is used to replace applicable.

Reviewer’s comment:

Line 81, “reported” is duplicated.

Authors response:

Duplication of the word reported has be corrected in text.

Reviewer’s comment:

Line 86, “Pure” is not at the right place in the sentence

Authors response:

Correction effected as “…were performed including a pure tone audiometric..”

Reviewer’s comment:

Line 143, you may want to precise “has 2 restrictions sites on the DNA amplified”

Authors response:

The corrected line now reads “The NciI restriction enzyme has 2 restriction sites on the DNA amplified (Figure 1A)…”

Reviewer’s comment:

Line 158, 162/163, please rephrase “homozygous normal (CC) with 163 p.Arg143=” so it reads “were not carrier of the variant”, or “non carrier of the variant”.

Authors response:

The line has been revised as “In the 8th family were two individuals affected by HI, one was heterozygous (CT) and the other one had the CC genotype (p.Arg143= )”

Reviewer’s comment:

Line 183 should read “Audiological characterization…”

Authors response:

 Correction made and the sentence now reads “Audiological characterization of participants ….”

Reviewer’s comment:

Figure 1.

Please indicate the limits of exon 2 of GJB2 as compared to the primers used for PCR and the restriction sites for NciI. Please replace the term “Mutant” by Homozygote carrier of the pathogenic variant. Please refer to the variant at both the DNA and the protein level, indicating the reference sequences, rather than just the protein change.

 Authors response:

The revision is made on the figure and legend and the variant has been referred to at the DNA level.

Reviewer’s comment:

Figure 2.

Please refer to the variant at both the DNA and the protein level. Please rephrase the legend.

Authors response:

The revision is made on the figure legend and the variant has been referred to at the DNA level.

Reviewer 3 Report

Several recessive pathogenic variants in GJB2 have been associated with profound hearing loss in Caucasian, Ashkenazi Jewish and East Asian populations. The presence of biallelic pathogenic variants in GJB2 is often a frequent cause of hearing loss in these populations. In Africa, this does not seem to be the case, except in Ghana where the pathogenic variant p.Arg143Trp in GJB2 is frequent. In this manuscript, the authors describe the design of a test, and its validation in 2 cohorts of patients with hearing loss and their associated controls, to identify the presence of the variant p.Arg143Trp in GJB2, which is frequent in Ghana. This test is based on a PCR targeting GJB2 only coding exon (exon 2), NciI restriction enzyme digestion of the PCR products, followed by gel electrophoresis and analysis. This variant p.Arg143Trp is pathogenic and when present as homozygote is associated with profound hearing loss. The authors show that this test is specific and sensitive and could thus be a very useful clinical test, which could be more easily implemented than classical sequencing approaches, as it does not require the use of a sequencer. The authors propose for this test to be done in addition to the new-born hearing screening program in Ghana.

The rationale for this study is strong.  The writing is generally clear although a few sentences require clarifications.  Some important updates regarding the role of GJB6 in hearing loss are needed. Although the authors’ conclusions are well supported by the data, presentation of a negative PCR control in figure 3 would be helpful (see below). Some clarifications are also needed relative to their hearing tests and the presentation of their audiometric results.

The implementation of the simple and robust test described in this work, in the clinical practice, could be very impactful for the early diagnosis of the genetic cause of hearing loss in Ghana.

Major comments:

1. Nomenclature of the variant should be clearly indicated at both the DNA and the protein level; reference accession numbers should be indicated.

2. The association of hearing loss and the presence of biallelic pathogenic variants in the coding sequence of GJB6 or of one variant in GJB6 coding sequence in association with a mutation in GJB2 (digenic inheritance), has been refuted (see Genet Med.2019 Oct;21(10):2239-2247. doi: 10.1038/s41436-019-0487-0. ClinGen expert clinical validity curation of 164 hearing loss gene-disease pairs. DiStefano MT et al. ClinGen Hearing Loss Clinical Domain Working Group.)

Please modify the text accordingly.

Deletions in the region of GJB2/GJB6, which are 2 small genes close to each other, are thought to be pathogenic through the modification of GJB2 expression. 

3. Presentation of a negative PCR control in figure 3 would be appreciated unless the authors prefer presenting the PCR results they initially obtained (with a negative control), before presenting their digestion results. Please present the results of Figure 3 as you suggest for the results to be presented in clinic. So that this article contains all the information necessary to do the test proposed, wouldn’t it be worth indicating the PCR conditions and the sequences of the primers used as template, still referencing the initial article those were first reported in as you did?

4. Please indicate clearly the results of which participants are included in Figure 4. Are these audiograms those of the 20 participants with hearing loss and 10 normal hearing family members from Adamorobe?

Please describe the material and method used to obtain the audiological data you present. Please explain what the pure tone average represents and the set of frequencies you chose to test and include. This should be explained in the text.

The pure tone average of the participants which carry the genotype TT are very high. Did you obtain some positive response in most of those patients? Is the material used capable of reliable detection of these high thresholds in the conditions of testing? 

Is there a difference in hearing thresholds between both ears in participants of each genotype?

Would it be better to present the data based on best ear and/or worse ear rather than left and right ear or even indicate both ears on the same graph and show the results at the different frequencies tested as a function of the genotype.

How different are those results as compared to those presented by G.W Brobby et al. 1998? Please comment

Please indicate in the legend of Figure 4 how many participants were tested in each group presented (TT, CT, CC). Please indicate their age or age range.

No comment is made in the manuscript regarding the age of the participants of each group. Do the patients with TT genotype present profound hearing loss from birth or is their hearing loss progressive?

Other comments:

The authors may want to replace along the manuscript “hearing impairment” by “hearing loss”, the preferred term in the field.

The use of the term “global” here along the manuscript seems inappropriate. As pointed out by the authors, the variant studied here is only found at a high frequency in Ghana and not in other countries of Africa. I would suggest taking this term out but keeping the rest of the sentences identical (lines 2, 35, 237).

Please include information on prevalence of GJB2 mutation in Ghana in the abstract, which support the importance of the diagnostic test described.

MIM number should be included in the text.

Line 18, “major” should be replaced by “most frequent”

Lines 58 to 61: please rephrase sentence

Line 64: you may want to change “applicable to” to “implementable in”

Line 81, “reported” is duplicated.

Line 86, “Pure” is not at the right place in the sentence

Line 143, you may want to precise “has 2 restrictions sites on the DNA amplified”

Line 158, 162/163, please rephrase “homozygous normal (CC) with 163 p.Arg143=” so it reads “were not carrier of the variant”, or “non carrier of the variant”.

Line 183 should read “Audiological characterization…”

Figure 1.

Please indicate the limits of exon 2 of GJB2 as compared to the primers used for PCR and the restriction sites for NciI. Please replace the term “Mutant” by Homozygote carrier of the pathogenic variant. Please refer to the variant at both the DNA and the protein level, indicating the reference sequences, rather than just the protein change.

Figure 2.

Please refer to the variant at both the DNA and the protein level. Please rephrase the legend.

Author Response

Reviewer 3:

Reviewer’s comment:

This manuscript reports on a RFLP test to detect the p.Arg1434Trp mutation in GJB2, which is a common cause of non-syndromic hearing impairment in the population of Ghana.

Reviewer’s comment:

Major comment

The manuscript describes a quick, cheap and easy method for the detection of a clinically relevant mutation. Additional details will improve the description. The NciI recognition sequence should be shown, and the nucleotide position that is affected by the mutation should be indicated. Note that the mutation destroys the recognition site and so the test is not theoretically specific. Any other change in the recognition site will be detected as well. For example, the c.426C>A change (resulting in the pathogenic variant p.Phe142Leu) would result in the same restriction pattern as that of p.Arg143Trp. Although the test will still be useful for genetic screening in Ghana, given the high frequency of p.Arg143Trp in this population, this issue should be discussed in the text, and it should be kept in mind when performing the screenings, i.e. Sanger confirmation would still be needed.

Authors response:

The restriction site of the enzyme has been added to the text of the method section as well as the legend of Figure 1. In the methods the sentence now read “Carefully selected restriction enzyme NciI (supplied by New England Biolabs Inc., USA, through Inqaba biotec™, South Africa) with the recognition site “CCSGG” was used to digest the PCR amplicons.”.

The legend of Figure 1 was edited to show the nucleotide position and it reads “Figure 1: NciI restriction fragment polymorphism investigations for GJB2-p.Arg143Trp (c.427C>T rs80338948) mutation. (A) Unipro UGENE [28] map of GJB2 exon 2 showing the primer binding sites (F. primer and R. Primer) and the restriction sites (CCSGG) for the restriction enzyme NciI and the resulting DNA fragments. (B) Expected gel electrophoresis result.”

The discussion has been expended to make room for other pathogenic variants that will show similar results as the founder mutation p.Arg134Trp. The line below was added to the discussion: “Nevertheless, the test cannot differentiate between other variants within the recognition site of the restriction enzyme; hence similar results would be obtained for the following pathogenic mutations: p.Phe142Leu (c.426C>A), p.Y142del (c.424_426delTTC), and p.Arg143Gln (c.428G>A).” To confirm the specific mutation at the enzyme restriction site, Sanger Sequencing would be needed.  However, the high prevalence of the GJB2-p.Arg143Trp mutation within the Ghanaian population makes the NciI-RFLP test very relevant.

Reviewer’s comment:

Minor points

In my opinion, the title is too long and the first sentence is a little pretentious. I would delete “Enhancing global Genetic Medicine:”. Line 81. “reported previously reported” (a typo) Line 99. “vinous blood” should read “venous blood” (a typo). Line 136. Did UNAFFECTED family members have moderate hearing impairment? I suppose that the authors mean that family members without the homozygous genotype had normal to moderate hearing impairment. Line 226. “profound severity of HI” should read “profound HI”.

Authors response:

The title has been revised as “Enhancing Genetic Medicine: Rapid and Cost-Effective Molecular Diagnosis for a GJB2 Founder Mutation for Hearing Impairment in Ghana”.

The typos “reported previously reported”, and “vinous blood” have been corrected in text.

Line 136 was edited to read “The unaffected family members without the homozygous TT genotype had normal to moderate hearing impairment”

Line 136 has been revised as “…that they had profound HI.”

Round 2

Reviewer 3 Report

Major comments

Thank you for your changes which significantly improved the manuscript.

Few points still need to be addressed:

As pointed out by the reviewer 3, and now more clearly stated in the text, this test although extremely valuable is not 100% specific. Please change the text accordingly all along the manuscript. Recognizing this does not change its usefulness in Ghana. You may want to indicate that no false positive was detected by this approach and that the sensitivity of the test was 100%. Please further rephrase the information relative to GJB6 you present along the manuscript taking into account the conclusions from the ClinGen expert panel reported by DiStefano et al. in Genetics in Medicine volume 21, pages2239–2247(2019) and cited below in case the authors don't have easy access to this article entitled "ClinGen expert clinical validity curation of 164 hearing loss gene-disease pairs": "One Refuted gene, GJB6, appeared on all 17 GTR panels examined, although this was not surprising. Coding variants in GJB6 are Definitively associated with Clouston syndrome/hidrotic ectodermal dysplasia, a syndrome characterized by hair loss and skin/nail abnormalities and no hearing loss, but its relationship with ARNSHL has only been documented through large genomic deletions, including GJB6-D13S1830 and GJB6-D18S1854. These deletions have been identified in trans with pathogenic GJB2 variants in many cases. Specifically, GJB6-D13S1830 is a deletion of approximately 309 kb of DNA including the 5′ end of GJB6 and a region upstream of both GJB6 and the GJB2 gene, which has been shown to eliminate a cis-acting element thereby abolishing expression of the cis-GJB2 allele. Additionally, an independent mouse model with only the coding sequence of GJB6 deleted and no surrounding sequence deleted had normal hearing, confirming that the regulatory region 5′ of GJB6, but not the gene itself, is necessary for normal hearing in mice. Furthermore, many deletions upstream of both GJB6 and GJB2 are pathogenic for hearing loss without disruption of GJB6. Therefore, the HL GCEP concluded that coding variants in GJB6 are not associated with hearing loss. "

Minor comments 

line 21 please replace "test for" by "test to detect" please remove or explain "strong evidence" line 101 of non environmental causes Figure 1: please indicate clearly the limits of exon 2 in the figure, as compared to the primers Figure 1 legend, key : please remove "unaffected/affected", you can have hearing impairment and not be a carrier of this pathogenic variant. Same comment for line 256 For the legend of Figure 1, please consider changing for the following: WT = Non carrier of the pathogenic variant (CC) MT = Homozygote for the pathogenic variant (TT) Het = Heterozygote for the pathogenic variant (CT) line 155 please indicate "impairment" instead of "impaired" line 160 please remove ("p.Arg143Arg") line 175 please remove "(p.Arg143=)" line 179 please change "homozygote normal (CC) with p.Arg143=" for "did not carry the pathogenic variant (CC)"

Author Response

Reviewer’s comment

Few points still need to be addressed:

As pointed out by the reviewer 3, and now more clearly stated in the text, this test although extremely valuable is not 100% specific. Please change the text accordingly all along the manuscript. Recognizing this does not change its usefulness in Ghana. You may want to indicate that no false positive was detected by this approach and that the sensitivity of the test was 100%. Please further rephrase the information relative to GJB6 you present along the manuscript taking into account the conclusions from the ClinGen expert panel reported by DiStefano et al. in Genetics in Medicine volume 21, pages2239–2247(2019) and cited below in case the authors don't have easy access to this article entitled "ClinGen expert clinical validity curation of 164 hearing loss gene-disease pairs": "One Refuted gene, GJB6, appeared on all 17 GTR panels examined, although this was not surprising. Coding variants in GJB6 are Definitively associated with Clouston syndrome/hidrotic ectodermal dysplasia, a syndrome characterized by hair loss and skin/nail abnormalities and no hearing loss, but its relationship with ARNSHL has only been documented through large genomic deletions, including GJB6-D13S1830 and GJB6-D18S1854. These deletions have been identified in trans with pathogenic GJB2 variants in many cases. Specifically, GJB6-D13S1830 is a deletion of approximately 309 kb of DNA including the 5′ end of GJB6 and a region upstream of both GJB6 and the GJB2 gene, which has been shown to eliminate a cis-acting element thereby abolishing expression of the cis-GJB2 allele. Additionally, an independent mouse model with only the coding sequence of GJB6 deleted and no surrounding sequence deleted had normal hearing, confirming that the regulatory region 5′ of GJB6, but not the gene itself, is necessary for normal hearing in mice. Furthermore, many deletions upstream of both GJB6 and GJB2 are pathogenic for hearing loss without disruption of GJB6. Therefore, the HL GCEP concluded that coding variants in GJB6 are not associated with hearing loss. "

Author’s response:

Thank you for providing us with the detailed breakdown of the conclusions of the paper. We have revised the manuscript based on your recommendations.

100% specific: The text has been edited based on the reviewer’s comments.

Reviewer’s comment

Minor comments 

line 21 please replace "test for" by "test to detect" please remove or explain "strong evidence" line 101 of non environmental causes Figure 1: please indicate clearly the limits of exon 2 in the figure, as compared to the primers Figure 1 legend, key : please remove "unaffected/affected", you can have hearing impairment and not be a carrier of this pathogenic variant. Same comment for line 256 For the legend of Figure 1, please consider changing for the following: WT = Non carrier of the pathogenic variant (CC) MT = Homozygote for the pathogenic variant (TT) Het = Heterozygote for the pathogenic variant (CT) line 155 please indicate "impairment" instead of "impaired" line 160 please remove ("p.Arg143Arg") line 175 please remove "(p.Arg143=)" line 179 please change "homozygote normal (CC) with p.Arg143

Author’s response:

Line 101: “strong” is removed from the sentence.

Line 21: "test for" was replaced with "test to detect"

The limits of exon 2 in the figure, as compared to the primers has been clearly shown in Figure 1.

The key: legend of Figure 1 and line 256 are revised accordingly.

Line 155: the word "impaired" was replaced with "impairment"

Line 160: ("p.Arg143Arg") has been removed

Line 175: "(p.Arg143=)" has been removed

line 179: "homozygote normal (CC) changed with p.Arg143
